# Liquid Biopsy as Novel Tool in Precision Medicine: Origins, Properties, Identification and Clinical Perspective of Cancer’s Biomarkers

**DOI:** 10.3390/diagnostics10040215

**Published:** 2020-04-13

**Authors:** Diego Fernández-Lázaro, Juan Luis García Hernández, Alberto Caballero García, Alfredo Córdova Martínez, Juan Mielgo-Ayuso, Juan Jesús Cruz-Hernández

**Affiliations:** 1Department of Cellular Biology, Histology and Pharmacology, Faculty of Health Sciences, University of Valladolid, Campus de Soria, 42003 Soria, Spain; 2Cancer Research Centre, University of Salamanca, 37007 Salamanca, Spain; jlgarcia@usal.es; 3Institute of Biomedical Research of Salamanca, IBSAL (University Hospital of Salamanca), 3007 Salamanca, Spain; jjcruz@usal.es; 4Department of Anatomy and Radiology, Faculty of Health Sciences, University of Valladolid, Campus of Soria, 42003 Soria, Spain; albcab@ah.uva.es; 5Department of Biochemistry and Physiology, Faculty of Health Sciences, University of Valladolid, Campus de Soria, 42003 Soria, Spain; a.cordova@bio.uva.es (A.C.M.); juanfrancisco.mielgo@uva.es (J.M.-A.); 6Oncology Service, University Hospital of Salamanca, 37007 Salamanca, Spain

**Keywords:** precision oncology, cancer, liquid biopsy, biofluids, circulating tumor cells, circulating tumor nucleic acids, exosomes, microvesicles

## Abstract

In recent years, there has been an increase in knowledge of cancer, accompanied by a technological development that gives rise to medical oncology. An instrument that allows the implementation of individualized therapeutic strategies is the liquid biopsy. Currently, it is the most innovative methodology in medical oncology. Its high potential as a tool for screening and early detection, the possibility of assessing the patient’s condition after diagnosis and relapse, as well as the effectiveness of real-time treatments in different types of cancer. Liquid biopsy is capable of overcoming the limitations of tissue biopsies. The elements that compose the liquid biopsy are circulating tumor cells, circulating tumor nucleic acids, free of cells or contained in exosomes, microvesicle and platelets. Liquid biopsy studies are performed on various biofluids extracted in a non-invasive way, and they can be performed both from the blood and in urine, saliva or cerebrospinal fluid. The development of genotyping techniques, using the elements that make up liquid biopsy, make it possible to detect mutations, intertumoral and intratumoral heterogeneity, and provide molecular information on cancer for application in medical oncology in an individualized way in different types of tumors. Therefore, liquid biopsy has the potential to change the way medical oncology could predict the course of the disease.

## 1. Introduction

Cancer is a spatial and temporal dynamic disease where differently evolving genetic clones are responsible for progression. In this landscape, the recognition of mechanisms responsible for tumor evolution remains a challenging task [1]. However, in recent years, there has been a notable increase in knowledge of cancer, accompanied by a very important technological development of highly sensitive molecular biology techniques which introduces us to the beginning of the application of precision medicine and particularly of precision oncology [2]. The main objective of precision oncology is to improve the diagnosis and treatment of cancer. To this end, a variety of genomic and molecular analyses can be applied to tumor material to help identify known predictive markers to guide the selection of treatment, derive a molecular subtype classification that might enable estimation of the prognosis, characterize somatic alterations involved in tumor progression, detect disrupted pathways and identify molecular discriminants of progression disease. However, access to tumor material for molecular profiling usually depends on invasive procedures that are not always feasible and do not lend themselves to serial monitoring of tumor genotypes [3].

Precision medical oncology in the clinical management of cancer may be achieved through the diagnostic platform called “liquid biopsy” (LB). The LB was recognized as a powerful real-time approach for the molecular monitoring of this dynamic cancer disease [4]. This method utilizes the detection of biomarkers in blood for prognostic and predictive purposes by non-invasive means, which in the near future will represent a change in the paradigm of molecular biology understanding and the heterogeneity of tumors [5]. The term LB was coined by Pantel and Alix-Panabières [6] to study circulating tumor cells (CTCs) in patients’ blood, but it is currently being extended to study cell-free circulating nucleic acids contained in exosomes and the information that platelets associated with tumors have. The LB’s studies may be performed on blood as well as other bodily fluids such as urine, saliva, cerebrospinal fluid (CSF) or pleural effusion, among others [6,7] (Figure 1).

## 2. Biofluids: Storage of Biomarkers

The selection of biofluid to obtain information about the disease will depend on the tumor and the accessibility of the sample. In the biofluids, there is molecular information provided by the genetic and epigenetic landscape, systematically following the tumor genomic evolution [7]. 

In malignancies with metastatic capacity such as breast, lung, colon or prostate cancers, once the primitive tumor has invaded the local extracellular matrices, its tumor cells will be able to migrate to distant locations in the body and establish secondary outbreaks, following different dissemination routes: direct, lymphatic and hematogenous. For this reason, CTCs and nucleic acids from the primary tumor can be found in the blood, thus blood is the most commonly used biofluid in the search for tumor biomarkers [6]. The obtaining of the sample, by means of blood drawn (simple and low-invasive technique), provides dynamic information on the progress and evolution of the type of cancer. 

In the plasma and/or serum, there are tumor marker proteins such as the carcinoembryonic antigen (CEA), the carbohydrate antigen 19-9 (CA19.9) or the prostate-specific antigen (PSA). In addition, the current revolution in the field of blood biomarkers makes it possible to study circulating nucleic acids, both circulating tumor DNA (ctDNA) and circulating tumor RNA (ctRNA), using highly sensitive genomic techniques [5].

Urine is divided into sediments, which allows the macroscopic study of crystalline structures in the form of salts and supernatants, where we find proteins, metabolites, nucleic acids and vesicles of extracellular origin [8]. The prostate-specific antigen (PSA) is found to increase in the urine in patients with prostate cancer, although the clinical routine determines it in blood [8]. Urine DNA comes from glomerular filtration where the fragments are 100 base pairs, although it depends on the patient’s condition. The enzymatic activity of DNAse-1 produces a high fragmentation of DNA with high (≥1 kpb) or low (<100 bp) molecular weight [9]. Urine DNA analysis has determined mutations in KRAS in pancreatic and colorectal cancers. In addition, in non-small-cell lung cancer, the epidermal growth factor receptor (EGFR) has detected mutations with sensitivities similar to those obtained in plasma. However, the detection of mRNA is not possible due to the action of RNases, although it is possible to detect micro RNA found inside urine exosomes [7].

In the saliva (hypotonic solution), there exist isolated proteins, DNA and RNA, and metabolites and microbiota, that are also present in the blood. Thus, their concentration changes can be used as biomarkers to detect early-stage cancer or to monitor the response to therapeutic management [10]. Salivary diagnostics are non-invasive, easy to use tools for patient specimen collection. Saliva testing potentially allows the patient to gather their own saliva samples, even at home, thus saving healthcare costs, enabling convenient and multiple sampling as well as having a positive impact on patient compliance [11].

In pancreatic cancer, the salivary messenger RNA can be used to detect KRAS, Methyl-CpG Binding Domain Protein 3-Like 2 (MBD3L2), Acrosomal Vesicle Protein 1 (ACRV1) and Dolichyl-Phosphate Mannosyltransferase Subunit 1 (DPM1) with 90% sensitivity and 95% specificity. All biomarkers together had greater diagnostic power than any single biomarker [12]. In addition, miRNA showed their importance in salivary diagnosis because they were expressed differentially in saliva samples from pancreatic cancer patients compared with controls [13,14]. Lau et al. [15] provide the role of exosomes derived from pancreatic cancer in the development of salivary biomarkers. In lung cancer, the salivary transcriptome was analyzed using mRNA as a biomarker and could differentiate lung cancer patients from control subjects [16,17]. In addition, some proteins such as haptoglobin hp2 (HP), zinc 2-glycoprotein (AZGP1) and calprotectin were differentially identified with high levels of sensitivity and specificity in the saliva of lung cancer patients [16,17]. Breast cancer expressed different biomarkers, similar to those expressed in saliva: epidermal growth factor (EGF), p53, human epidermal growth factor receptor 2 (HER2/neu) and carbohydrate antigen 15-3 (CA 15-3) [10]. Salivary proteins for the detection of gastric cancer were detected as biomarkers, discriminating between healthy controls and patients [18]. Finally, the quantity and quality of salivary DNA is similar to plasma and has been used to detect mutations of phosphatidylinositol 3-kinase (PI3K), the cyclin-dependent inhibitor of kinase 2A (CDKN2A), F-box and WD repeat domain-containing 7 (FBXW7), HRAS and KRAS, with a 100% detection sensitivity with tumors located in the oral cavity [19].

Although obtaining CSF involves an invasive procedure, it is a better alternative in tumors located in the central nervous system where most of the patients analyzed had circulating nucleic acids in CSF but not in the plasma [6]. The purpose of this blood test is for enumerating CTCs. CTCs are cancer cells that detach from a primary tumor and travel through the bloodstream or lymphatic system to other parts of the body an LB that can be used at any time during a patient’s course of the disease [6,7].

## 3. Comparison of Tissue Biopsy and Liquid Biopsy

Tissue biopsy has traditionally been considered a key tool for the diagnosis and control of numerous diseases. In the case of cancer, tissue biopsy makes it possible to determine the histological properties of the tumor, as well as its genetic profile, in order to diagnose and predict its evolution and even predict the response to treatment. Until recently, tissue was the only way to study tumor mutations. Despite advances in obtaining tissue samples, biopsies present real limitations such as the difficulty of performing them due to the location itself, the absence of visible tumor and the implicit iatrogenic of the method and, finally, due to the impossibility of repeating it frequently, which rules it out as a monitoring method. This is common, for example, in lung cancer, mainly in stages I or II, which are very early stages [20]. Other limitations may be due to patient comorbidities, logistical availability, cost and, in the case of analyzing tissues stored in banks, the elapsed time that can degrade many elements. However, the great restriction is determined by the evolutionary nature of the type of cancer itself since, in the biological process of formation and expansion of the tumor mass, the clones diverge and form different subpopulations or sub-clones that cause tumor heterogeneity. This diversity and plasticity are not represented in a tissue sample, limited in time and space. In this sense, one of the tools that make it possible to overcome these barriers is LB (Figure 2) [21].

## 4. Liquid Biopsy Components

The components of LB mainly refer to the analysis of CTCs, cell-free circulating nucleic acids, mainly ctDNA or ctRNA, exosomes, microvesicle and platelets. All of them are released into the peripheral blood of the primary tumor and/or metastatic deposits [22].

### 4.1. Circulating Tumor Cells (CTCs)

#### 4.1.1. General Characteristics

CTCs are cancer cells that actively or passively depart from the primary tumor mass following multiple potential patterns of dissemination to distant localization and, as a consequence, adaptation to different tissue microenvironments [23]. CTCs are exceptionally rare, heterogeneous blood elements that include a mixture of sub-populations with dissimilar phenotypic and functional features [24]. The most used cytopathological definition of CTC has been set: a cell with a round or oval morphology, a visible round or oval nucleus within the cytoplasm and CTCs are distinguished from other blood cells by their positive expression of epithelial markers, including epithelial cell adhesion molecules (EpCAM) and cytokeratins (CK) and negative expression of the white blood cell-specific marker CD45 [23,25]. Heterogeneous CTCs can be divided into various subgroups: intact single CTCs, apoptotic CTCs and CTCs clusters. The size range of CTCs (8–20 µm) is slightly greater than that of other blood cells, but the size is also heterogeneous and can be similar or smaller than surrounding white blood cells (WBCs) [26].

#### 4.1.2. Origin: Epithelial to Mesenchymal Transition (EMT)

The Epithelial to Mesenchymal Transition (EMT) is a process composed of different phases in which an epithelial cell acquires a mesenchymal phenotype. Among the changes that occur are loss of cell polarity, acquisition of a migratory capacity, invasiveness, resistance to apoptosis and increased production of extracellular matrix components [27]. All these changes occur as a consequence of the activation and repression of genes involved in specific signaling pathways related to this event. There are growth factors as inducers of EMT, including transforming growth factor β (TGF-β), epidermal growth factor (EGF), Wnt, Snail/Slug, Twist and Six1 [6].

In addition, it involves a large number of signaling pathways that act independently or interconnected, most of which converge in the control of E-cadherin expression, whose under-regulation is the key molecular event in this process, since it causes the loss of intercellular junctions and destabilization of epithelial architecture. It consists of three phases: a phase of non-migratory cells, pre-migratory cells and migratory cells (TGF-B, Snail, TWIST, Sox, Slug, ZEB1, among others), and proteins involved (E-cadherin, integrin, vimentin, occlusins and claudins) [27,28]. EMT occurs in three different biological contexts, with different functional consequences: (i) Type 1 EMT associated with developmental processes; (ii) Type 2 EMT as the ultimate survival mechanism of cancer cells associated with healing and tissue regeneration processes; (iii) Type 3 EMT associated with tumor progression. Type 3 EMT consists of the exchange of cancer cells of epithelial origin for tumor cells with mesenchymal characteristics capable of invading and spreading, leading to the formation of metastases [29].

In this line, which concerns the mechanisms that initiate EMT, it has been proposed that the genetic and epigenetic alterations suffered by the tumor cells during the formation of the primary tumor make them respond to EMT-inducing signals from the tumor-associated stroma [30]. The origin and the processes involved in CTCs dissemination are from EMT. EMT is a phenomenon involved in embryogenesis and wound healing processes, which might be triggered during cancer progression and metastasis. In cancer cells, EMT may lead to a significant change in cell phenotype, associated with a more aggressive biological behavior, inducing the loss of both cell junctions and apical–basal polarity, enabling cell motility and assisting CTCs during intravasation into the bloodstream. This phenomenon, and its opposite process, the Mesenchymal to Epithelial Transition (MET), could enable CTCs to switch backward and forward from the epithelial to the mesenchymal phenotype, also causing resistance to anoikis, to the physical stress induced by blood circulation and to chemo- and radiotherapy [23,24,26]. Thus, CTCs in the bloodstream have heterogeneous levels of biomarker expression related to EMT and MET [31].

Thus, EMT facilitates the metastatic spread and progression of tumor cells from the primary tumor to surrounding tissues and distant organs. Expression changes that take place in cells during the EMT process alter the polarity of epithelial cells so that they acquire the morphological and biochemical traits of mesenchymal cells. This process is essential for tumor cells to elude apoptosis, anoikis, cell senescence and to escape the immune system response. For all these reasons, it is important to bear in mind that tumor cells modify their expression throughout the metastatic process [6]. 

#### 4.1.3. Methods: Isolation, Identification and Culture

CTCs are extremely rare and extremely limited in numbers in comparison to other types of blood cells (1–10 CTCs/10^6^ blood cells in 1 mL of blood). Even if 99.9% of blood cells are purified, the purity of the isolated CTCs is below 1%, which is at the detection limit of the currently available molecular analysis techniques [5,6]. However, methodological limitations in recent technological development have allowed much progress in methods to isolate CTCs from blood, taking into account the heterogeneity of the CTCs in relation to surface protein markers and physical characteristics. Therefore, before isolation, to consider the specificity and sensitivity, and the tests that are desired to be performed in CTCs such as quantification, molecular characterization or functional studies are required to improve the clinical utility of LB [32].

On the morphological, biological and physical properties, the isolation methods used are: (i) direct visualization of cellular morphology for the identification of malignant phenotypes. This is an important and unique aspect of CTCs in comparison to other circulating biomarkers and other blood cells. In addition, immunocytochemistry (ICC) and fluorescent in situ hybridization (FISH) can be performed on CTCs to identify their phenotype and detect some genomic alteration. CTCs can be divided into various subgroups, according to the cell diameter or nuclear fraction, and each subgroup is analyzed for its characteristics [33,34]; (ii) immune-isolation, using specific antibodies that identify molecules present on the cell surface of CTCs; (iii) physical properties of CTCs exclusion by size, density, deformability or cell dielectric point [35]; (iv) based on positive and negative enrichment of the blood fraction obtained. Positive enrichment allows a positive selection of CTCs, uses antigens such as EpCAM (expressed in epithelial cells) or specific organ markers such as HER2, EGFR and CEA. However, CTCs isolation technologies that use anti-EpCAM antibodies cannot isolate cells that have undergone EMT or have low EpCAM expression. With respect to negative enrichment, the unselected fraction is the one that interests the most commonly used antigen, which is CD45 [32].

Actually, technological advances have made it possible to develop different methods for the determination of CTCs in clinical laboratories: (i) Morphological (Isolation by Size of Epithelial Tumor cells (ISET)); (ii) Based on density gradient (Oncoquick); (iii) Immunomagnetic (Magnetic Activated Cell Sorting system (MACS^®^), AdnaTest^®^, RosetteSep-Applied imaging Rare Event (RARETM^®^), Fiber-optic Array Scanning Technology (FAST^®^), Laser Scanning Cytometer (LSC^®^), CellSearch^®^, CellCollector^®^). In addition, microscopes are available automated systems that allow for further exploration (Automated Cellular Imaging System (ACIS^®^), CellSpotter and ARIOL^®^) [36].

Emerging immune-affinity technology is the CellSearch^®^ system, a reference diagnostic platform for isolation and detection of CTCs using immune-magnetic technology. It provides an accurate and reproducible analysis that allows CTCs to be detected at a density as low as CTCs in 7.5 mL of whole blood, with a specificity of >99% and 97% sensitivity. This technology has been approved by the Food Drug American (FDA) for breast, colorectal and prostate cancer testing. This blood test for enumerating CTCs. CTCs are cancer cells that detach from a primary tumor and travel through the bloodstream or lymphatic system to other parts of the body. An LB that can be used at any time during a patient’s course of disease [35,37]. Methodologically, this system uses whole blood and bases the isolation of CTCs on the expression of positive EpCAM using magnetic beads. It then performs immunofluorescence of the enriched fraction of CTCs to identify those that are positive for cytokeratin and negative for CD45. The system itself proposes events as potential CTCs and it is the specialist technician who must select those that meet the CTCs+ criteria (which combines marker expression and morphology, the nucleus/cytoplasm ratio). The current CellSearch^®^ system defines a CTC as an event that has a nucleus (DAPI positive), expresses cytokeratin (CK8, CK18 and CK19), does not express CD45 and is more than 4 × 4 µm^2^ in size [38]. Therefore, the CellSearch^®^ system CTC test is a simple, practical blood test that helps oncologists evaluate the prognosis of patients with metastatic, prostate, or colorectal breast cancer [39]. 

A novel instrument by immune-affinity technology is CellCollector^®^ (Gilupi). This tool is the first in vivo CTCs isolation device reported to overcome the limitations of small blood sample volumes in the entire world that has CE approval. It is placed directly in the torrent A patient’s blood through an indwelling catheter (size = 20 G), remains in the vein of the arm for 30 min and allows the capture of CTCs in vivo since the system is in contact with a high volume of blood. The catheter is lined with anti-EpCAM antibody and subsequently allows the characterization of CTCs (immunofluorescence, protein analysis, RT–qPCR, FISH, sequencing, analysis of expression and cell culture) [40,41]. While being safe and easy to use, the CellCollector^®^ did not outperform CellSearch^®^ in terms of CTCs yield or sensitivity. While CTC detection in cancer colorectal patients in baseline was significantly increased with the CellCollector^®^, the clinical relevance of these CTCs appears inferior to the cells identified by the CellSearch^®^ system [42]. For CellCollector^®^, the sensitivity and specificity of the in vivo CTC detection strategy for the diagnosis of early-stage lung cancer were 52.94% and 90%, respectively [41]. In addition, for CellCollector^®^, although the sensitivity of CTC detection needs to be further increased, may suggest that high CTC counts might contribute to the identification of high-risk prostate cancer patients with occult metastases at the time of diagnosis [40].

The isolation methodology described by Pantel et al. [43] establishes that it is obligatory that the CTCs maintain their integrity and are viable in order to be able to carry out functional tests. The isolation techniques to be used must not establish a fixation phase or produce cellular stress. In addition, for in vitro CTCs cultures, sterile conditions should be used. In this sense, the EPithelial Immuno SPOT (EPISPOT) is a system that allows analysis of the secretion of isolated CTCs, which has identified viable CTC-detecting KRT-19, MUC1, PSA and FGF-2 secreted [44].

Isolated CTCs are cultured in vitro to establish the permanent CTC cell line for further studies. Furthermore, they can be used in various in vivo, ex vivo, and in vitro experiments in functional studies. CTC-derived explants (CDXs), patient-derived xenograft models (PDXs) and ex vivo-cultured CTCs can be used in drug screening, which can contribute to personalized therapy [26]. However, there was still a methodological limitation is the low number of cells isolated per sample, because not all of them are viable and the short half-cell life (2 h reaching a maximum of 24 h since the migration of the primary tumor) [38].

#### 4.1.4. Potential Application of Circulating Tumor Cells (CTCs) in Medical Oncology

Recently has described a lot of evidence that the number of CTCs can be 1000 or more cells in 1 mL of blood [45,46]. However, Gold et al. [47] have reported an average metastatic carcinoma patient has between 5 and 50 CTCs for approximately every 7.5 mL of blood (<1 to >5 CTCs/mL). Probably, the heterogeneity of the CTCs in blood could be the reason for not generating metastasis [46]. The CTCs may potentially contain valuable information on tumor composition, invasiveness, drug susceptibility and resistance to the therapy used [7]. Therefore, the determination of prognostic importance of CTC detection on patients with cancer, shows that a large number of CTCs is associated with poor treatment response, shortening of the overall, disease-free survival rates, aggressive disease, increased metastasis and decreased relapse time [48]. CTCs are biomarkers of LB potentially capable of guiding therapeutic management, determining the efficacy or need for therapy, even in the absence of detectable metastases, and providing information on the mechanisms of drug resistance [6,7,43].

In summary, the potential application of CTCs in medical oncology in all stages of cancer disease: diagnosis, efficacy of treatment, selection of pharmacological targets, emergence of resistance and compression of minimal residual disease (MRD) [35,49]. Although CTCs are important biomarkers in LB, their detection, quantification and isolation are still being improved for the purpose of being clinically validated in different types of cancer in the near future [32].

### 4.2. Cell-Free Circulating DNA (cfDNA)

Cell-free circulating nucleic acids can be passively released by apoptotic and necrotic cells, or actively released by living cells. CTCs contain various cellular and subcellular components that can be used for further downstream analysis; e.g., intact DNA for well-known mutation analysis and novel marker discovery, RNA for gene expression profiling and various biomarkers for proteomic analysis. In addition to CTCs, another major component in the blood is nucleic acids and especially the fraction derived from tumor cells ctDNA and ctRNA [26]. The nucleic acids, consisting of both cell-free circulating DNA (cfDNA) and cell-free circulating RNA (cfRNA) was first described more than 60 years ago and in the blood of cancer patients and was first recognized in the 1970s when high levels of cfDNA were detected in the serum of cancer patients. Since then, cfDNA has found applications in many disciplines of medicine such as evaluation of fetal DNA [32]. 

However, not all cfDNA found in blood is of tumor origin and may come from blood cells or other tissues [50]. In healthy individuals, it mostly originates from apoptosis of blood nucleated cells found at low levels in blood plasma and may increase in some inflammatory conditions, after exercise and benign breast diseases [32]. Patients with malignant tumors had higher cfDNA levels than those with benign tumors and these levels are highly elevated cfDNA, particularly in advanced disease stages, with a wide range of 0%–65%, in part due to reduced DNAse activity [38]. The necrosis process is not the main source of circulating DNA, although it can contribute to its levels since fragments of DNA between 21 and 80 kb are observed that coincide with the fragments released by the excision of chromatin in nucleosomes, by the action of deoxyribonuclease activated by caspases, which are activated during apoptosis cell-free circulating nucleic acids can be actively released through vesicles secreted by living cells, such as exosomes and microvesicles [9].

In order to extract cfDNA from the blood, different methods have been developed: (i) Magnetic enrichment of cfDNA can be achieved by positively charged magnetic beads that bind the negatively charged phosphate backbone of DNA, may be more efficient than other conventional methods in the recovery of short cfDNA fragments; (ii) Silica column-based enrichment makes use of the binding affinity of DNA molecules, with high recovery of 82%–92% cfDNA from serum but a low yield and partial loss of DNA fragments smaller than 150 bp [51]. Finally, the molecular biology techniques used in LB could be used to detect the specific genetic alterations as potential targets for targeted cancer therapy because the analysis of cfDNA or cfRNA (especially the fraction derived from tumor cells) enables the detection of specific genetic alterations [52].

#### 4.2.1. Circulating Tumor DNA (ctDNA)

ctDNA is single- or double-stranded DNA, and exists in plasma or serum. Early studies showed that ctDNA possessed many cancer-associated molecular characteristics, such as single-nucleotide mutations, methylation changes and cancer-derived sequences, and therefore were considered to be derived from tumor tissue [53].

It is hypothesized that ctDNA is secreted as a signaling molecule to drive tumor metastasis by oncogenic transformation of susceptible cells. There are three potential origins of ctDNA: (i) Apoptotic or necrotic tumor cells; (ii) Live tumor cells; (iii) CTCs [54]. In addition, there are two processes involved in the release of ctDNA into the blood circulating: (i) Passive release of DNA through cell death either by apoptosis or necrosis. As a consequence of enzymatic cleavage of DNA during apoptosis, the resulting DNA fragments are still wrapped around single nucleosomes and the length plus linker is around 166 bp. Larger fragments starting from 320 bp, the length of DNA wrapped around two nucleosomes, up to N1000 bp are released from phagocytosis of necrotic cells; (ii) active secretion ctDNA takes place by the release of extracellular vesicles, such as exosomes and prostasomes, containing pieces of DNA around 150–250 bp [51]. The rate of ctDNA shedding into the circulation depends on the location, size, and vascularity of the tumor, leading to a difference in ctDNA levels 0.01%–90% found in blood. The half-life time of ctDNA in the blood circulation ranges from 16 min to 2.5 h. The concentration of the total cfDNA in healthy individuals is on average 30 ng/mL plasma and ranges from 0 to 100 ng/mL, whereas, in cancer patients, this can be up to 1000 ng/mL [51,53].

#### 4.2.2. Methodologies for Detection of ctDNA

The average amount of cfDNA in 1 mL of plasma from patients with advanced-stage cancer (17 ng) and the amount of DNA contained by a single human cell (6 peak grams) would require more than 2 × 10^3^ CTCs releasing their genetic content in that volume of plasma. However, only 10 CTCs are detected on average in 7.5 mL of blood. This fact, together with its high concentration in the blood, places cfDNA as the good biomarker of the tumor present in LB [44]. However, cfDNA is highly fragmented DNA and the total amount of ctDNA might make up as low as 0.01% of the total cfDNA. These extremely low concentrations make the detection challenging, particularly at the early stages of tumor development [55].

In this way, some molecular biology techniques: Q-PCR, BEAMing, Safe-SeqS, CAPP-Seq and TAmSeq use one or a few known tumor-specific mutations of the primary tumor to monitor residual disease in peripheral blood allows targeted monitoring can be extremely sensitive, as mutations can be detected at an allele frequency of up to 0.01% with high specificity and at a fast and cost-effective speed. Although, it requires detailed information on the tumor genome [56]. Other techniques for investigating ctDNA involve unfocused screening and aims at the genome-wide analysis of copy number aberrations (CNAs) or point mutations by whole-genome sequencing (WGS) or whole-exome sequencing (WES). These techniques have the ability to identify new changes that occur during treatment of the tumor and no prior information about the genome of the primary tumor is required but high concentrations of ctDNA are required and low sensitivity (5%–10%) [57].

Recently, digital polymerase chain reaction (dPCR) has been used to yield an absolute measure of nucleic acid concentrations. A new method referred to as droplet digital PCR (ddPCR) has gained attention as a more precise and less subjective assay to quantify DNA amplification. It has been proven the usefulness of ddPCR to determine HER2 gene amplification of breast cancer. The ddPCR can be used as a molecular analysis tool to precisely measure HER2 copy number alterations in formalin-fixed paraffin-embedded (FFPE) samples of heterogeneous breast tumors at both the RNA and DNA levels [58].

#### 4.2.3. Potential Application of ctDNA in Medical Oncology

ctDNA is a biomarker for liquid biopsy in cancer diagnosis and prognosis. LB based on ctDNA is superior to that of previous plasma biomarkers in two main areas: (i) sensitivity; (ii) clinical correlations. For ctDNA analysis, plasma samples are preferable to serum samples to avoid possible contamination of wild-type DNA released from other blood cells. The unfavorable stability of ctDNA is advantageous for real-time monitoring of the disease status, although it adversely affects the sensitivity of the detection. The half-life of ctDNA is less than 2 h, whereas the half-life of protein markers in plasma can be several weeks. This means that ctDNA can more accurately reflect the real-time tumor burden in patients receiving therapy. Indeed, ctDNA has been shown to correlate well with tumor load and the likelihood of recurrence. Furthermore, ctDNA can originate from any tumor lesion in the body, so sampling bias is minimized. Thus, ctDNA offers an advantage for application in medical oncology [32,57,59].

Other applications of ctDNA are genotyping and assistance in personalized, targeted therapy. Genotyping is aimed at the analysis of genetic mutations and has important applications in managing cancer treatment which would be of significant benefit to the field of personalized medicine. The ctDNA analysis overcomes tissue biopsy problems by reflecting the genetic mutations of the whole tumor tissue. Additionally, ctDNA from the same patients at different stages can be used to dynamically monitor the genetic mutations during the cancer progression [53].

Therefore, tumor cfDNA is more heterogeneous because it will derive from a combination of apoptosis, necrosis and active secretion of proliferating cancer cells, CTCs, and the renewal of tumor cells disseminated in cell tissues with micrometastasis. The heterogeneity of cfDNA, together with the limitation of the classical biopsy that only represents a small tumoral region, allows positioning of the LB through the repetition of blood sampling as a potential method for application in medical oncology in all stages of cancer disease [38].

### 4.3. Cell-Free Circulating Free RNA (cfRNA)

Although less abundant than cfDNA, also there is cell-free circulating (cfRNA) that can be determined in the serum of patients. This cfRNA can be found: ctRNA, microRNA (miRNA), messenger RNA (mRNA) and non-coding RNAs and for monitoring tumor genetic dynamics [35]. 

#### 4.3.1. Circulating Tumor Messenger RNA 

ctRNA in blood is also used to identify mutations in genes of interest, but additionally, provides information about relative expression levels of important genes. In this sense, programmed cell death-1 ligand-1 (PD-L1) overexpression in cancer cells accelerates tumor progression. PD-L1 possesses two main pro-oncogenic functions. First, PD-L1 is a strong immunosuppressive molecule that inactivates tumor-specific T cells by binding to the inhibitory receptor PD-1. Second, PD-L1 function relies on the delivery of intrinsic intracellular signals that enhance cancer cell survival, regulate stress responses and confer resistance toward pro-apoptotic stimuli, such as interferons [60]. PD-L1 expression was detected in the plasma ctRNA of all cancer types at varying frequencies but no PD-L1 messenger RNA (mRNA) was detected in cancer-free individuals there was a high degree of concordance between expression of PD-L1 protein in tumor tissues and PD-L1 gene expression in plasma, and both methods were equally predictive of pharmacological response. These findings may provide additional predictive information on the outcome of patients on anti-PD-L1 therapy [61]. In addition, elevated levels of telomerase reverse transcriptase mRNA (hTERT) are often found in different types of tumors such as breast or colon cancer but no hTERT mRNA was detected in cancer-free individuals [59]. However, there are difficulties in using mRNA as a biomarker in LB because mRNA is protected in extracellular vesicles, and is associated with protein complexes. Additionally, the reproducibility and utility of cell-free mRNA as a biomarker are limited due to its instability, low abundance, and intracellular mRNA contamination from specimen processing [62].

#### 4.3.2. Circulating Tumor MicroRNAs 

In the last years, several studies have focused on microRNAs as a tumor suppressor or oncogenic functions can provide the cancer cells with a proliferative advantage and promote metastasis [63]. The microRNAs (miRNAs) are released by cells either in association with RNA-binding proteins or packaged inside exosomes, which protect these cell-free miRNAs against RNase activity [13]. In addition, the concentration of miRNAs in blood circulation and other biofluids is higher than mRNA and more stable in cancer patients [59]. The miRNAs expression levels have been proven to be potentially valuable for the early diagnosis, prognosis, and prediction of the response to therapy in various types of cancer using multiple analyses of miRNAs [63]. In non-small cell lung cancer were found the following miRNAs: MiR-145, miR-20a, miR-21, and miR-223 overexpressed in plasma, miR-10b and miR-373 detect lymph node metastases in breast cancer patients [64,65].

Before analyzing miRNA, it is necessary to use isolation methods which allow them to be separated from the contaminants (RNA’s coagulation process, platelets, cell debris, associated with proteins, inside exosomes), which represent the major drawbacks of using cell-free miRNAs as biomarkers in cancer [7]. The methods of analysis that provide more information on miRNAs can be detected by three well-established and sensitive methods: qRT–PCR, microarray and deep sequencing [62]. The qRT–PCR has an acceptable sensitivity and needs a low concentration of RNA, although it needs a standard curve and its specificity is limited to the design of the “primer” [7].

### 4.4. Exosomes

Exosomes are 40- to 150-nm extracellular vesicles of endosomal origin secreted by all types of cells, functioning as mediators of intercellular communication. The exosomal cargo, including proteins (cytoskeletal, transmembrane and thermal shock proteins), lipids, enzymes (GAPDH, ATPase, PGK1) and nucleic acids mainly mRNA and miRNA and single-stranded and two-stranded DNA. The content of exosomes is dependent on the cell of origin and can be transferred into adjacent or distant recipient cells, modulating their intracellular signaling pathways, gene expression, and phenotype [66]. Thus tumor-derived exosomes reflect the nature and status of the original tumor cells because they contain all cellular contents as the tumor cell origin. Furthermore, an abundant number of exosomes is released by tumor cells in comparison with non-tumorigenic cells, found in most body fluids, such as blood, serum, urine, cerebral spinal fluid, and even breast milk [62].

The most commonly used method for the isolation of exosomes entails a series of successive centrifugation and ultracentrifugation steps that allow the acquisition of a final sample free of cells and other cell contaminants. The purity of the final sample is also influenced by the centrifugation speed, duration, temperature and the rotor used. The magnetic beads coated with antibodies against exosomal markers, such as CD9 or CD63, can also be used to isolate exosomes used with small sample volumes and with the aim of isolating specific populations [67].

There are different platforms to facilitate the use of exosomes as diagnostic and prognostic biomarkers in cancer as: (i) ExoScreen is a highly sensitive and rapid analytical technique for profiling circulating EVs/exosomes directly from blood samples of patients has also been established that uses two types of antibodies to capture two specific antigens residing on EVs and can be detected by photosensitizer beads. Using this system, higher-signal-detecting CD147 and CD9 double-positive EVs were observed in serum from colorectal cancer (CRC) patients compared with that in serum from healthy donors. A part of sera from CRC patients including stage I and II were positive for CD147, although both CEA and CA19-9 positive serum could not be found in those patients [68]; (ii) ExoTest found an increase in plasma exosomes expressing CD63 and caveolin-1 in melanoma patients; (iii) ExoQuick is an efficient method for the isolation of exosomes compared to ultracentrifugation and is suitable to obtain a good yield of exosomal miRNA an mRNA from serum and urine, as exosomal miR-191, miR-21, and miR-451a as potential early diagnostic markers in pancreatic cancer [62]. The detection and quantification of exosomes that permit their use as specifically cancer markers can be detected by ELISA, nanoparticle tracking analysis, flow cytometry and transmission electron microscopy [67].

#### Potential Application of Tumor-Derived Exosomes in Medical Oncology

The LB by developing exosomic profiles in the absence of tissue biopsies could have very promising results for the early diagnosis of cancer and its monitoring to evaluate the efficacy of antitumor therapy. In this sense, exosomes play an important role in the different stages of development in breast cancer through the horizontal transfer of genetic information between their cells that include stimulation of tumor angiogenesis, reorganization of the stroma to establish the tumor microenvironment, as well as promoting tumor growth and drug resistance [5]. In pancreatic ductal adenocarcinoma (PDAC) patients KRASG12D, TP53R273H mutations can be found in circulating exosomal DNA and were identified as glypican 1 (GPC1)-positive serum exosomes. In addition, in breast cancer, GPC1 patients exhibited high levels. Epidermal growth factor variant III (EGFRvIII) and miRNA-21 was also found to be increased in serum exosomes tumor derived from glioblastoma patients. In addition to MET, the proteins tyrosinase-related protein-2 (TYRP2), very late antigen 4 (VLA-4), heat-shock protein 70 (HSP70) and HSP90 can be found increased in circulating TEXs in patients with advanced melanoma [62].

### 4.5. Microvesicles

The microvesicles are larger than the exosomes (100–1000 nm); the composition of their membrane is more similar to that of the cell from which they come in such a way that they practically represent the proteomic profile of the tumor. Microvesicles arise from the membrane of different types of cells: hematopoietic cells, endothelial cells, mesenchymal stem cells and cancer cells. Throughout the transfer of their molecular contents, microvesicles are capable of altering the function of recipient cells. Increasing evidence suggests a key role for microvesicles mediated intercellular communication in a variety of cellular processes involved in tumor development and progression, including immune suppression, angiogenesis and metastasis. Aspects of microvesicles biogenesis or function are therefore increasingly being considered as targets for anti-cancer therapy. In breast cancer, tumor-associated macrophages secrete microvesicles with miRNAs inside that stimulate the aggressiveness of cancer cells [69].

### 4.6. Platelet

The possibility of using blood platelets altered by a tumor as a biomarker in LB is recent and platelet tumor education can reflect the tumor profile and may serve as an informative tool in clinical cancer diagnosis. The blood platelets and cancer cells interact on the sequestration of tumor-associated biomolecules by platelets, resulting in their “education” and affects not only the expression of relevant genes in tumor cells, but also alters the RNA profile of blood platelets. Moreover, platelets are capable of taking up circulating mRNAs and proteins from the tumor microenvironment and can also sequester extracellular vesicles from cancer cells harboring tumor-specific RNA [70]. In platelets of metastatic lung cancer patients compared to healthy individuals, there are differential splice isoforms of NAD-dependent deacetylase sirtuin-2 (SIRT2). In addition, in various cancer types as glioblastoma, non-small cell lung cancer, colorectal cancer, pancreatic cancer, breast cancer and hepatobiliary was identified an mRNA sequencing of platelet tumor educated from 283 platelet samples cancer patients from healthy individuals with 96% accuracy [62].

## 5. Applications of Liquid Biopsy’s Biomarkers

The LB’s biomarkers are a reality in clinical application and others are very promising tools in the near future. LB has been shown to be effective for its application in different types of tumors including lung, colorectal, prostate, melanoma, breast and pancreatic cancer. In lung cancer, LB would detect the expression of PD-L1 in CTCs or in white blood cells, although with the limitation of the isolation of these CTCs and the concordance with tissue, and the clinical impact of the same. Actually, the immunotherapy based on antibodies against the Programmed Death-ligand 1 (PD-L1) and epidermal growth factor receptor (EGFR) tyrosine kinase inhibitors are some targeted therapies in lung cancer [71]. In addition, in colorectal cancer has been described the genotyping studies establish a concordance of more than 90% between KRAS tests in tumor tissue and ctDNA, making it possible to test in ctDNA LB as valid for tissue tests in patients with colorectal cancer and may be applied in the proximate clinic [72]. One of the most important variants of the receptors is AR variant 7 (AR-V7), a truncated form of the same, whose existence has been associated with more aggressive prostate tumors resistant to castration and whose level of expression is related to the risk of recurrence after radical prostatectomy. It is also associated with resistance to enzalutamide and abiraterone, which could make it a marker for treatment selection [73]. In this sense, Garcia et al. [73] have described for the first time, the feasibility of detecting AR-V7 expression in proteins present in the plasma of prostate cancer patients at different stages by means of an automated capillary nano-immunoassay “WES^TM^” (ProteinSimple, San Jose, CA, USA), however, it is still only one promising biomarker that is not used in the clinic. In melanoma is one promising biomarker the predictive value of ctDNA, in patients with metastatic melanoma, by quantitative analysis of plasma mutations of BRAF V600 has unique properties as a monitoring tool during treatment with BRAF/MEK inhibitors, both as a predictor of response and treatment resistance [74,75]. LB based on ctDNA detection, such as digital PCR (ddPCR) assays, have demonstrated the clinical utility of ddPCR in determining HER2 mutations in breast cancer [38,58,76]. Finally, in pancreatic cancer, Cohen et al. have shown that the analysis of KRAS gene mutations, using ctDNA, had greater diagnostic power than any single marker as carcinoembryonic antigen (CEA); carbohydrate antigen 19-9 (CA19-9); cancer antigen 125 (CA125) but it’s not yet a reality in clinical practice [77].

## 6. Conclusions

LB represents the analysis of tumor-derived material in the blood and other body fluids of cancer patients. The components of LB mainly refer to the analysis of CTCs, cell-free circulating nucleic acids, exosomes, microvesicles and platelets. These elements that make up the range of tumor components in the blood, molecular analysis is possible that provides different and complementary information (Table 1) and have shown to reflect the genetic profile of both primary and metastatic lesions and provide real-time monitoring of tumor dynamics, representing a great promise for personalized medicine. Its importance is because LB can be performed in different biofluids and allows to overcome the limitations of tissue biopsy as the recurrent problems in the clinical assessment of tumors that stem from the lack of accessibility to the tumor tissue and its clonal heterogeneity. LB is a complementary or alternative method for tissue biopsy. This portrays a minimally invasive detection tool for molecular biomarkers offering possibilities of tumor identification reflecting in real-time the tumor heterogeneity.

To our mind, this is important because LB can be performed in different biofluids (blood, urine, saliva, cerebral, CSF and pleural effusion) and allows to overcome the limitations of tissue biopsy as the recurrent problems in the clinical assessment of tumors that stem from the lack of accessibility to the tumor tissue and its clonal and it could show in real-time the tumor heterogeneity. In addition to allowing it to be repeated as many times as necessary, it could establish a molecular profile that opens the door to a significant number of clinical applications that are the target of precision medicine. 

However, we believe, although the scientific, instrumental, human and technological challenge is ambitious with the current advances it is possible that it will be achieved in the immediate future. Subsequently, it will be necessary to validate and standardize the entire process, from obtaining the sample, through its analysis to demonstrating its clinical usefulness. At present, the question arises as to whether the results obtained in the biofluids reproduce those obtained in tissue. Probably, the limitations of the LB that prevent it from being implanted as a routine technique or from being clinically valid is the previous description that allows us to choose the biomarker source according to the objective. Another important limitation is the technical complexity of liquid biopsy, as it is so complex that it is not found in day hospitals with a large flow of patients, and this fact prevents large-scale clinical trials. For these reasons, the introduction of LB, in clinical practice could be a tool complementary or alternative method to tissue biopsy.

## Figures and Tables

**Figure 1 diagnostics-10-00215-f001:**
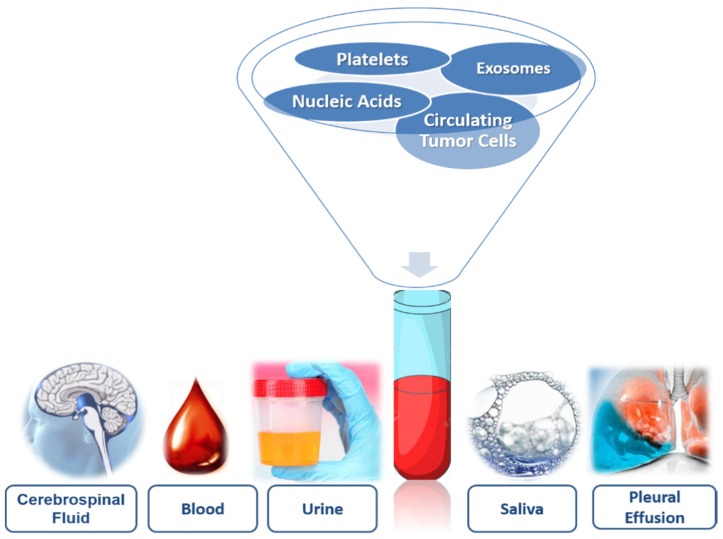
Components of liquid biopsy.

**Figure 2 diagnostics-10-00215-f002:**
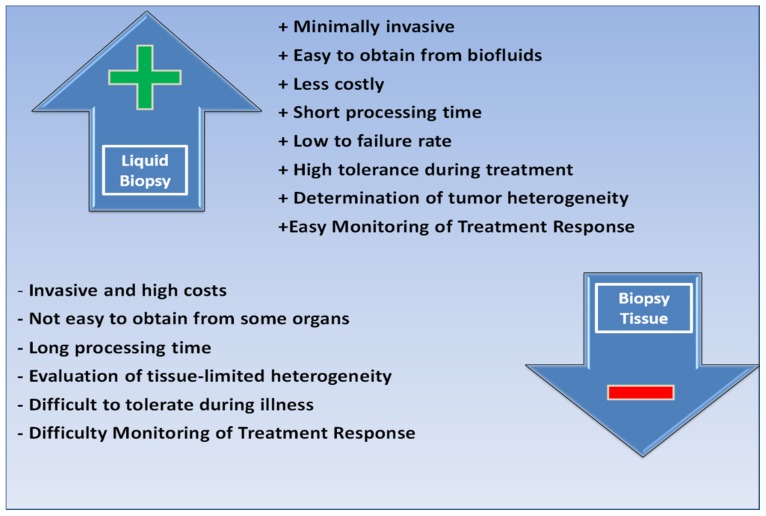
Comparison of tissue biopsy and liquid biopsy.

**Table 1 diagnostics-10-00215-t001:** Applications of the elements of circulating tumor cells (CTCs), tumor nucleic acids (ctDNA/ctRNA) and exosomes. Adapted from Siravegna et al. (2018) [7].

Elements	CTCs	ctDNA/ctRNA	Exosomes
Potential to assess tumor spatial and temporal heterogeneity	No	Yes	No
Detection of somatic mutations, indels, alteration of the number of copies and gene fusions	Yes	Yes	Si
Analysis of mRNA, miRNA, IncRNA and gene variants	Yes	Yes	Yes
Evaluation of methylation patterns	Yes	Yes	Yes
Methylation Studies	Yes	No	Yes
Expression Studies	Yes	No	Si
Cell morphology and functional studies ex vivo	Yes	No	No
Signal co-location demonstration	Yes	No	No
Proteomic analysis	Yes	No	Yes

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
