# Peer review of "Liquid Biopsy as Novel Tool in Precision Medicine: Origins, Properties, Identification and Clinical Perspective of Cancer’s Biomarkers"

_diagnostics, 2020, doi:10.3390/diagnostics10040215_

Round 1
Reviewer 1 Report
Remarks:
- The review is poorly structured.The authors do not indicate what problems exist for the widespread use of liquid biopsy in a clinical practice. Of the various methods and approaches for detecting CTCs, only CELLSEARCH® and EPISPOT are listed. There is no data on the sensitivity and specificity of detection methods for CTCs. Section 1.2. Origin: Epithelial to Mesenchymal Transition (EMT) is very poorly represented. The inclusion of the section “Comparison of tissue biopsy and liquid biopsy” would improve this review.
- . The article has many typos and errors. The authors point to Figure 1 (line 60), while the figure itself is not given (it is not in the article).
….The BL´s studies may be performed on blood as well as other body fluids such as urine, saliva, cerebrospinal fluid (CSF) or pleural effusion, among others [6, 7] (Figure 1)…..
Literature references are not appropriate.
For example, in Table 1 (Table 1. Applications of the elements of circulating tumor cells (CTCs), tumor nucleic acids (ctDNA / ctRNA) and exoxomas. Adapted from Siravegna et al. (2018) [8].), The authors cite the work Siravegna et al. (2018) [8].
Although under number 8 in the list of the references is another author.
The authors write that "The metastatic carcinoma shows between 5-50 CTCs in approximately 7.5 ml of blood." Although there is a lot of evidence that the number of CTCs can be 1000 or more cells in 1 ml of blood (Clinical Chemistry, Volume 65, Issue 4, 1 April 2019, Pages 549–558, https://doi.org/10.1373/ clinchem. 2018.296814; Molecules. 2018 Mar 22; 23 (4). pii: E727. doi: 10.3390 / molecules23040727.)
- In conclusion, the authors do not have their own opinions about what problems need to be addressed for the widespread introduction of liquid biopsy in clinical practice.
Author Response
Point-by-Point Response to Reviewer’s Comments
We would like to sincerely thank the reviewer for his/her helpful recommendations again. We have seriously considered all the comments and carefully revised the manuscript accordingly. Revisions are highlighted in yellow through the manuscript to indicate where changes have taken place. We feel that the quality of the manuscript has been significantly improved with these modifications and improvements based on the reviewers’ suggestions and comments. We hope our revision will lead to an acceptance of our manuscript for publication Diagnostics.
Also, the manuscript has undergone English language editing by a Professor at the Faculty of Translation, University of Valladolid. The text has been checked for correct use of grammar and common technical terms, and edited to a level suitable for reporting research in a scholarly journal.
In advance,
King regards
Rev: The authors do not have their own opinions about what problems need to be addressed for the widespread introduction of liquid biopsy in clinical practice.
Authors: Thank you for your recommendation. Authors have included two paragraph with our opinion in Conclusion section.
Conclusions
To our mind, this is important because LB can be performed in different biofluids (blood, urine, saliva, cerebral, CSF and pleural effusion) and allows to overcome the limitations of tissue biopsy as the recurrent problems in the clinical assessment of tumors that stem from the lack of accessibility to the tumor tissue and its clonal and it could show in real time the tumor heterogeneity. In addition to allowing it to be repeated as many times as necessary, it could establish a molecular profile that opens the door to a significant number of clinical applications that are the target of precision medicine.
However, we believe, although the scientific, instrumental, human and technological challenge is ambitious with the current advances it is possible that it will be achieved in the immediate future. Subsequently, it will be necessary to validate and standardize the entire process, from obtaining the sample, through its analysis to demonstrating its clinical usefulness. At present, the question arises as to whether the results obtained in the biofluids reproduce those obtained in tissue. Probably, the limitations of the LB that prevent it from being implanted as a routine technique or from being clinically valid is the previous description that allows us to choose the biomarker source according to the objective. Another important limitation is the technical complexity of liquid biopsy, as it is so complex that it is not found in day hospitals with a large flow of patients, and this fact prevents large-scale clinical trials. For these reasons the introduction of LB, in clinical practice could be a tool complementary or alternative method to tissue biopsy.
Rev: Of the various methods and approaches for detecting CTCs, only CELLSEARCH® and EPISPOT are listed. There is no data on the sensitivity and specificity of detection methods for CTCs.
Authors: Thank you for your observation. Authors have included a new information
- Even if 99.9% of blood cells are purified, the purity of the isolated CTCs is below 1%, which is at the detection limit of the currently available molecular analysis techniques [5, 6].
- Actually, technological advances have made it possible to develop different methods for the determination of CTCs in clinical laboratories: i) Morphological (Isolation by Size of Epithelial Tumor cells (ISET)); ii) Based on density gradient (Oncoquick); iii) Immunomagnetic (Magnetic Activated Cell Sorting system (MACS®), AdnaTest®, RosetteSep-Applied imaging Rare Event (RARETM®), Fiber-optic Array Scanning Technology (FAST®), Laser Scanning Cytometer (LSC®), CellSearch®, CellCollector®). In addition, microscopes are available automated systems that allow for further exploration (Automated Cellular Imaging System (ACIS®), CellSpotter and ARIOL®) [36].
- CellSearch®: It provides an accurate and reproducible analysis that allows CTCs to be detected at a density as low as CTCs in 7.5 mL of whole blood, with a specificity of >99% and 97% sensitivity.
- A novel instrument by immune-affinity technology is CellCollector® (Gilupi). This tool is the first in vivo CTCs isolation device reported to overcome the limitations of small blood sample volumes in the entire world that has CE approval. It is placed directly in the torrent A patient's blood through an indwelling catheter (size 20G), remains in the vein of the arm for 30 minutes and allows the capture of CTCs in vivo since the system is in contact with a high volume of blood. The catheter is lined with anti-EpCAM antibody and subsequently allows the characterization of CTCs (immunofluorescence, protein analysis, RT-qPCR, FISH, sequencing, analysis of expression and cell culture) [40, 41]. While being safe and easy to use, the CellCollector® did not outperform CellSearch® in terms of CTCs yield or sensitivity. While CTCs detection in cancer colorectal patients in baseline was significantly increased with the CellCollector®, the clinical relevance of these CTCs appears inferior to the cells identified by the CellSearch® system [42]. For CellCollector® the sensitivity and specificity of the in vivo CTCs detection strategy for the diagnosis of early-stage lung cancer were 52.94% and 90%, respectively [41]. Also, CellCollector® although the sensitivity of CTC detection needs to be further increased, may suggest that high CTC counts might contribute to the identification of high-risk prostate cancer patients with occult metastases at the time of diagnosis [40].
Rev: Section 1.2. Origin: Epithelial to Mesenchymal Transition (EMT) is very poorly represented.
Authors: Thank you for your observation. Authors have included a new information in Origin: Epithelial to Mesenchymal Transition (EMT) section
Origin: Epithelial to Mesenchymal Transition (EMT)
The Epithelial to Mesenchymal Transition (EMT) is a process composed of different phases in which an epithelial cell acquires a mesenchymal phenotype. Among the changes that occur are: loss of cell polarity, acquisition of a migratory capacity, invasiveness, resistance to apoptosis and increased production of extracellular matrix components [27]. All these changes occur as a consequence of the activation and repression of genes involved in specific signaling pathways related to this event. There are growth factors as inducers of EMT, including transforming growth factor ß (TGF-ß), epidermal growth factor (EGF), Wnt, Snail/Slug, Twist and Six1 [6].
In addition, it involves a large number of signaling pathways that act independently or interconnected, most of which converge in the control of E-cadherin expression, whose under-regulation is the key molecular event in this process, since it causes the loss of intercellular junctions and destabilization of epithelial architecture. It consists of three phases: A phase of non-migratory cells, pre-migratory cells and migratory cells; (TGF-B, Snail, TWIST, Sox, Slug, ZEB1 among others), and proteins involved (E-cadherin, integrin, vimentin, occlusins, and claudins) [27, 28]. EMT occurs in three different biological contexts, with different functional consequences: i) Type 1 EMT associated with developmental processes; ii) Type 2 EMT as the ultimate survival mechanism of cancer cells. associated with healing and tissue regeneration processes; iii) Type 3 EMT associated with tumor progression. Type 3 EMT consists of the exchange of cancer cells of epithelial origin for tumor cells with mesenchymal characteristics capable of invading and spreading, leading to the formation of metastases [29].
In this line, the mechanisms that initiate EMT, it has been proposed that the genetic and epigenetic alterations suffered by the tumor cells during the formation of the primary tumor make them respond to EMT-inducing signals from the tumor-associated stroma (31). The origin and the processes involved in CTCs dissemination is EMT. EMT is a phenomenon involved in embryogenesis and wound healing processes, which might be triggered during cancer progression and metastasis. In cancer cells, EMT may lead to a significant change in cell phenotype, associated with a more aggressive biological behavior, inducing the loss of both cell junctions and apical-basal polarity, enabling cell motility, and assisting CTCs during intravasation into the bloodstream. This phenomenon and its opposite process, the Mesenchymal to Epithelial Transition (MET), could enable CTCs to switch backward and forward from the epithelial to the mesenchymal phenotype, also causing resistance to anoikis, to the physical stress induced by blood circulation and to chemo- and radiotherapy [23, 24, 26]. Thus, CTCs in the bloodstream have heterogeneous levels of biomarker expression related to EMT and MET [31].
Thus, EMT facilitates the metastatic spread and progression of tumor cells from the primary tumor to surrounding tissues and distant organs. Expression changes that take place in cells during the EMT process alter the polarity of epithelial cells so that they acquire the morphological and biochemical traits of mesenchymal cells. This process is essential for tumor cells to elude apoptosis, anoikis, cell senescence and to escape the immune system response. For all these reasons, it is important to bear in mind that tumor cells modify their expression throughout the metastatic process [6].
Rev: The review is poorly structured. The authors do not indicate what problems exist for the widespread use of liquid biopsy in a clinical practice. The inclusion of the section “Comparison of tissue biopsy and liquid biopsy” would improve this review.
Authors: Thank you for your observation. Authors have included a new section that include both point.
Comparison of tissue biopsy and liquid biopsy
Tissue biopsy has traditionally been considered a key tool for the diagnosis and control of numerous diseases. In the case of cancer, tissue biopsy makes it possible to determine the histological properties of the tumor, as well as its genetic profile in order to diagnose and predict its evolution and even predict the response to treatment. Until recently, tissue was the only way to study tumor mutations. Despite advances in obtaining tissue samples, biopsies present real limitations such as the difficulty of performing them due to the location itself, the absence of visible tumor and the implicit iatrogenic of the method, and finally due to the impossibility of repeating it frequently, which rules it out as a monitoring method. This is common, for example, in lung cancer, mainly in stages I or II, which are very early stages [20]. Other limitations may be due to patient co-morbidities, logistical availability, cost and, in the case of analyzing tissues stored in banks, the elapsed time that can degrade many elements. But the great restriction is determined by the evolutionary nature of the cancer itself since, in the biological process of formation and expansion of the tumor mass, the clones diverge and form different subpopulations or sub-clones that cause tumor heterogeneity. This diversity and plasticity are not represented in a tissue sample, limited in time and space. In this sense, one of the tools that make it possible to overcome these barriers is the LB (Figure 2) [21].
Figure 2. Comparison of tissue biopsy and liquid biopsy
Rev: The authors point to Figure 1 (line 60), while the figure itself is not given (it is not in the article).….
Authors: Thank you for your observation. Authors have included a figure.
Figure 1. Components of Liquid Biopsy
Rev: Literature references are not appropriate. For example, in Table 1 (Table 1. Applications of the elements of circulating tumor cells (CTCs), tumor nucleic acids (ctDNA / ctRNA) and exoxomas. Adapted from Siravegna et al. (2018) [8].), The authors cite the work Siravegna et al. (2018) [8]. Although under number 8 in the list of the references is another author.
Authors: Thank you for your recommendation. The authors, have corrected the error.
Table 1. Applications of the elements of circulating tumor cells (CTCs), tumor nucleic acids (ctDNA/ctRNA) and exoxomes. Adapted from Siravegna et al. (2018) [7].
Rev: The authors write that "The metastatic carcinoma shows between 5-50 CTCs in approximately 7.5 ml of blood." Although there is a lot of evidence that the number of CTCs can be 1000 or more cells in 1 ml of blood.
Authors: Thank you for your observation. Authors have included a new paragraph.
Recently has described a lot of evidence that the number of CTCs can be 1000 or more cells in 1 ml of blood [45, 46]. However, Gold et al. [47] have reported an average metastatic carcinoma patient has between 5 and 50 CTCs for approximately every 7.5 mL of blood (<1 to >5 CTCs/mL). Probably, the heterogeneity of the CTCs in blood could be the reason for not generating metastasis [46]. The CTCs may potentially contain valuable information on tumor composition, invasiveness, drug susceptibility, and resistance to the therapy used [7]. Therefore, the determination of prognostic importance of CTCs detection on patients with cancer, shows that a large number of CTCs is associated with poor treatment response, shortening of the overall, disease-free survival rates, aggressive disease, increased metastasis and decreased relapse time [48].

Reviewer 2 Report
The authors present a nice review into the components and potential utilities of liquid biopsies. I like the focus on all the different cell and biomarker types, but the manuscript needs significant english editing in order to flow better and make more sense.
I think the biggest improvement to this manuscript would be to clarify which liquid biopsy cell types, biomarkers, or methods are already being used in the clinic, which ones are being evaluated in clinical trials, and which ones are just 'promising'. For example, EGFR resistance mutation detection in lung cancer.
A figure depicting how all the different mentioned liquid biopsy components work together could be a nice addition as well.
3.2: No mention of droplet-based methods like ddPCR which are widely used in certain cancer types for diagnosis, detection, monitoring using ctDNA.
section 3.3: cfRNA is not 'circulating free DNA' but 'cell-free DNA' or 'cell-free circulating RNA.' Also should lines 312-317, which are about cfNA be in the section 3.2? seems out of place.
3.3.1: Please discuss more how ctRNA PD-L1 expression correlates with IHC, as only 'high degree of concordance' was mentioned. Is this true of all tumor types?
Lots of spelling errors: like cfARN instead of cfRNA, or cfAN instead of cfNA, lncARN instead of lncRNA, biposia instead of biopsy, BL instead of LB, exoxomas instead of exosomes.
There are a lot of grammar errors, and I have hard time understanding and following the narrative, even as a native english speaker. Only a few examples given below:
line 41: "important technological development that introduces us to the path of precision medicine and in particular oncology." I'm not sure if the authors mean by technological development. Liquid biopsies? NGS?
line 96: "messenger RNA allows the of KRAS .." Do the authors mean "messenger RNA can be used to detect .."
line 105: "with respect of breast cancer were expressed differentially..." what does this mean?
line 116: "Patients diagnosed with local lymphomas in the nervous system were identified as miRNA" -- what does this mean??
line 182: "permits to list" what does this mean?
line 204: "however, up to this day did the low number of cells isolated per sample" what does this mean?
Author Response
Point-by-Point Response to Reviewer’s Comments
We would like to sincerely thank the reviewer for his/her helpful recommendations again. We have seriously considered all the comments and carefully revised the manuscript accordingly. Revisions are highlighted in green through the manuscript to indicate where changes have taken place. We feel that the quality of the manuscript has been significantly improved with these modifications and improvements based on the reviewers’ suggestions and comments. We hope our revision will lead to an acceptance of our manuscript for publication Diagnostics.
Also, the manuscript has undergone English language editing by a Professor of English at the Faculty of Translation, University of Valladolid. The text has been checked for correct use of grammar and common technical terms, and edited to a level suitable for reporting research in a scholarly journal.
In advance,
King regards
Rev: Authors have written about liquid biopsy cell types, biomarkers, or methods are already being used in the clinic, which ones are being evaluated in clinical trials, and which ones are just 'promising'
Authors: Thank you for your recommendation. For the reason, we have included a new section: Applications of liquid biopsy´s biomarkers
Applications of liquid biopsy´s biomarkers
The LB´s biomarkers are a reality in clinical application and others are very promising tools in the near future. LB has been shown to be effective for its application in different types of tumors including lung, colorectal, prostate, melanoma, breast and pancreatic cancer. In lung cancer, LB would detect the expression of PD-L1 in CTCs or in white blood cells, although with the limitation of the isolation of these CTCs and the concordance with tissue, and the clinical impact of the same. Actually, the immunotherapy based on antibodies against the Programmed Death-ligand 1 (PD-L1) and epidermal growth factor receptor (EGFR) tyrosine kinase inhibitors are some targeted therapies in lung cancer [71]. Also, in colorectal cancer has been described the genotyping studies establish a concordance of more than 90% between KRAS tests in tumor tissue and ctDNA, making it possible to test in ctDNA LB as valid for tissue tests in patients with colorectal cancer and may be applied in the proximate clinic [61]. One of the most important variants of the receptors is AR variant 7 (AR-V7), a truncated form of the same, whose existence has been associated with more aggressive prostate tumors resistant to castration and whose level of expression is related to the risk of recurrence after radical prostatectomy. It is also associated with resistance to enzalutamide and abiraterone, which could make it a marker for treatment selection [73]. In this sense, Garcia et al. [73] have described for the first time, the feasibility of detecting AR-V7 expression in proteins present in the plasma of prostate cancer patients at different stages by means of an automated capillary nano-immunoassay '”WESTM” (ProteinSimple, San Jose, CA, USA), however, it is still only one promising biomarker that is not used in the clinic. In melanoma is one promising biomarker the predictive value of ctDNA, in patients with metastatic melanoma, by quantitative analysis of plasma mutations of BRAF V600 has unique properties as a monitoring tool during treatment with BRAF/MEK inhibitors, both as a predictor of response and treatment resistance [74, 75]. LB based on ctDNA detection, such as digital PCR (ddPCR) assays, have demonstrated the clinical utility of ddPCR in determining HER2 mutations in breast cancer [38, 59, 76]. Finally, in pancreatic cancer, Cohen et al. have shown that the analysis of KRAS gene mutations, using ctDNA, had greater diagnostic power than any single marker as carcinoembryonic antigen (CEA); carbohydrate antigen 19-9 (CA19-9); cancer antigen 125 (CA125) but it's not yet a reality in clinical practice [77].
Rev: A figure depicting how all the different mentioned liquid biopsy components work together could be a nice addition as well.
Authors: Thank you for your observation. Authors have included a figure that mentioned liquid biopsy.
Figure 1. Components of Liquid Biopsy
Rev: No mention of droplet-based methods like ddPCR which are widely used in certain cancer types for diagnosis, detection, monitoring using ctDNA.
Authors: Thank you for your observation. Authors have included a new paragraph.
Recently, digital polymerase chain reaction (dPCR) has been used to yield an absolute measure of nucleic acid concentrations. A new method referred to as droplet digital PCR (ddPCR) has gained attention as a more precise and less subjective assay to quantify DNA amplification. It has been proven the usefulness of ddPCR to determine HER2 gene amplification of breast cancer. The ddPCR can be used as a molecular-analysis tool to precisely measure HER2 copy number alterations in formalin-fixed paraffin-embedded (FFPE) samples of heterogeneous breast tumors at both the RNA and DNA levels [59].
Rev: section 3.3: cfRNA is not 'circulating free DNA' but 'cell-free DNA' or 'cell-free circulating RNA.' Also should lines 312-317, which are about cfNA be in the section 3.2? seems out of place.
Authors: Thank you for your observation. Authors have re-written.
Lines 119-120: The components of LB mainly refer to the analysis of CTCs, cell-free circulating nucleic acids of mainly ctDNA or ctRNA, exosomes, microvesicle and platelets. All of them are released into the peripheral blood of the primary tumor and/or metastatic deposits [22].
Line 221 section 3.2 Authors have corrected the spelling errors.
3.2. Cell-free circulating DNA (cfDNA)
The nucleic acids, consisting of both cell-free circulating DNA (cfDNA) and cell-free circulating RNA (cfRNA), was first described more than 60 years ago and in the blood of cancer patients and was first recognized in the 1970s when high levels of cfDNA were detected in the serum of cancer patients. Since then, cfDNA has found applications in many disciplines of medicine such as evaluation of fetal DNA [32].
Rev: line 41 "important technological development that introduces us to the path of precision medicine and in particular oncology." I'm not sure if the authors mean by technological development. Liquid biopsies? NGS?
Authors: Thank you for your recommendation. Authors have rewritten the phrase.
Lines 40-42: However, in recent years there has been a notable increase in knowledge of cancer, accompanied by a very important technological development of highly sensitive molecular biology techniques which introduces us to the beginning of the application of precision medicine and particularly of precision oncology [2].
Rev: line 96: "messenger RNA allows the of KRAS .." Do the authors mean "messenger RNA can be used to detect .."
Authors: Thank you for your observation. Authors have rewritten the phrase.
In pancreatic cancer, the salivary messenger RNA can be used to detect: KRAS, Methyl-CpG Binding Domain Protein 3 Like 2 (MBD3L2), Acrosomal Vesicle Protein 1 (ACRV1) and Dolichyl-Phosphate Mannosyltransferase Subunit 1 (DPM1) with 90% sensitivity and 95% specificity. All biomarkers together had greater diagnostic power than any single biomarker [12].
Rev: line 105: "with respect of breast cancer were expressed differentially..." what does this mean?
Authors: Thank you for your observation. Authors have corrected the errors.
Regarding breast cancer were expressed different biomarkers were expressed in saliva: epidermal growth factor (EGF), p53, human epidermal growth factor receptor 2 (HER2/neu) and carbohydrate antigen 15-3 (CA 15-3) [10].
Rev: line 116: "Patients diagnosed with local lymphomas in the nervous system were identified as miRNA" -- what does this mean??
Authors: Thank you for your recommendation. Authors have rewritten the phrase.
In this way, miRNA (15b, 21, 19, 92a) was determinate in patients with local lymphomas in the nervous system [7].
Rev: line 182: "permits to list" what does this mean?
Authors: Thank you for your recommendation. Authors have rewritten the phrase.
This blood test for enumerating CTCs. CTCs are cancer cells that detach from a primary tumor and travel through the bloodstream or lymphatic system to other parts of the body. A LB that can be used at any time during a patient's course of disease. [29, 30].
Rev: line 204: "however, up to this day did the low number of cells isolated per sample" what does this mean?
Authors: Thank you for your recommendation. Authors have rewritten the phrase
However, there was still methodological limitation technological development in recent years has allowed much progress in methods to isolate CTCs from blood, taking into account the heterogeneity of the CTCs in relation to surface protein markers and physical characteristics. Therefore, before isolation, to consider the specificity and sensitivity, and the tests that are desired to be performed in CTCs such as quantification, molecular characterization or functional studies are required to improve the clinical utility of LB [32].

Round 2
Reviewer 1 Report
Authors have seriously considered all the comments and carefully revised the manuscript accordingly. I feel that the quality of the manuscript has been significantly improved with these modifications. I recommendat to accept this manuscript in present form.
Reviewer 2 Report
Overall, I like the changes. I still think the grammar needs review by a native English speaker.